# A Comprehensive Benchmark for Neural Human Radiance Fields

**Kenkun Liu[1,2*], Derong Jin[2], Ailing Zeng[1†], Xiaoguang Han[2], Lei Zhang[1]**
[1]International Digital Economy Academy (IDEA)
[2]The Chinese University of Hong Kong, Shenzhen
`https://kenkunliu.github.io/HMNeRFBench/`

## Abstract

The past two years have witnessed a significant increase in interest concerning NeRF-based human body rendering. While this surge has propelled considerable advancements, it has also led to an influx of methods and datasets. This explosion complicates experimental settings and makes fair comparisons challenging. In this work, we design and execute thorough studies into unified evaluation settings and metrics to establish a fair and reasonable benchmark for human NeRF models. To reveal the effects of extant models, we benchmark them against diverse and hard scenes. Additionally, we construct a cross-subject benchmark pre-trained on large-scale datasets to assess generalizable methods. Finally, we analyze the essential components for animatability and generalizability, and make HumanNeRF from monocular videos generalizable, as the inaugural baseline. We hope these benchmarks and analyses could serve the community.

## 1 Introduction

The free-view human body rendering and animation have gained significant attention due to their broad applications in industries such as film-making, video games, metaverse, and AR/VR. Recently, the neural radiance field (NeRF) [33] provides a new neural implicit representation that employs a multi-layer perceptron (MLP) to encode object density and view-dependent color for each point in a 3D scene, which has been widely adopted for human body rendering. The vanilla NeRF requires a batch of multi-view images for training to represent a single static scene, which is far from real applications. Thus, some works [35, 23, 42, 58, 53] attempt to reduce the number of views while some other works [46, 7, 30, 22, 9] try to adapt the per-scene training setting to be one-shot training. Furthermore, there are also some works [39, 15, 50, 5, 44, 27] trying to model dynamic scenes.

The extended NeRFs for humans [38, 49, 12, 54, 10, 37, 32, 8, 24] follow similar pathways of development. Specifically, there have developed two streams of methods for human body rendering, i.e., scene-specific methods and generalizable methods. The former aims to encode a human's high-fidelity appearance with as few train views and video frames as possible and then synthesize the human image in novel views or novel poses. One of the representative works is NeuralBody [38], which requires four views and 100-300 frames for training. Another work, HumanNeRF [49], takes a step further with only monocular video frames for training. It exploits the prior of human body shape and deformation, making the single-view frames competitive to multi-view images. Recently, generalizable methods have tried to train a universal model that can be directly used to synthesize a novel-view human image. Like NHP [24], it only needs three views as input to render the human's image in any given view. Additionally, to achieve novel pose rendering, MPSNeRF [16] first warps observed appearance information to canonical space and then to the target pose space.

---

*Work done during an internship at IDEA.

†Corresponding author.

Table 1: Comparisons of recent NeRF-based human rendering methods on different aspects. In the column **Dataset**, **ZM**, **PS**, **GB**, **HM**, **H36M**, **RP** are ZJU-MoCap [38], People-Snapshot [1], GeneBody [11], HuMMan [4], Human3.6M [19], RenderPeople datasets, respectively. **Estimated** means to use estimated masks and SMPL parameters instead of the ground-truth. **Views**: train views for scene-specific methods and input views (*) for generalizable methods. **Frames**: train frames for scene-specific methods and input frames (*) for generalizable methods.

| Method | Dataset | Views | Frames | Estimated | Generalizable | Animatable | Unified Evaluation |
|---|---|---|---|---|---|---|---|
| NeuralBody [38] | ZM, PS | 4 | 100-300 | ✗ | ✗ | ✓ | ✗ |
| AniNeRF [37] | ZM, H36M | 4 | 100-300 | ✗ | ✗ | ✓ | ✗ |
| Arah [47] | ZM, H36M | 4 | 300-400 | ✗ | ✗ | ✓ | ✗ |
| HumanNeRF [49] | ZM | 1 | 500-600 | ✗ | ✗ | ✓ | ✗ |
| UV-Volume [10] | ZM, H36M | 18 | 100 | ✗ | ✗ | ✓ | ✗ |
| MonoHuman [54] | ZM | 1 | 500-600 | ✗ | ✗ | ✓ | ✗ |
| NHP [24] | ZM, AIST++ | 3* | 1* or 3* | ✗ | ✓ | ✗ | ✗ |
| MPSNerf [16] | ZM, H36M, THuman | 3* | 1* | ✗ | ✓ | ✓ | ✗ |
| GP-NeRF [8] | ZM | 3* | 1* | ✗ | ✓ | ✗ | ✗ |
| KeypointNeRF [32] | ZM | 3* | 1* | ✗ | ✓ | ✗ | ✗ |
| GNR [11] | ZM, GB, RP | 4* | 1* | ✗ | ✓ | ✗ | ✗ |
| Ours | ZM, GB, HM | 1 or 4, 1* | 1, 60, 100, 300, 500, 10* | ✓ | ✓ | ✓ | ✓ |

While these methods progress greatly, some emerging problems should be taken seriously. First, existing methods are evaluated on different datasets, metrics, and settings (e.g., used views, frames, ground-truth masks, and SMPLs [31]), making systematic comparison hard. For instance, generalizable methods NHP and GP-NeRF have different train and test splits. Detailed comparisons are listed in Tab. 1. There is a lack of comprehensive ablation studies about the number of train views and frames that will influence the results. Second, existing commonly used dataset (e.g., ZJU-MoCap [38]), which contains a small number of actors performing easy actions wearing simple close-fitting clothes, is hard to reflect the effectiveness of these methods in real scenarios. Although other datasets (including Human3.6M [19], People-Snapshot [1], AIST++ [26], THuman [57] and RenderPeople) are also used in some works, they are still not complex, diverse, and large-scale enough. Third, for generalizable methods, the train data scale is too small (e.g., on ZJU-MoCap). Fourth, given monocular videos, there is currently no exploration of achieving both generalizability and animatability simultaneously.

To address the above issues, this work builds a new benchmark for human NeRF models, where 1) we establish a comprehensive benchmark for human NeRF models from unified evaluation metrics and experimental settings and retrain the representative methods to serve future work for fair comparisons; 2) we elaborately explore challenging datasets (e.g., GeneBody [11] and HuMMan [4]), which contain large-scale subjects with diverse actions and challenging clothes, and conduct comprehensive experiments to evaluate existing methods. Instead of averaging the quantitative results on all subjects, we classify the selected dataset into several categories to separately evaluate the existing methods' performance on different typical cases; 3) we build a benchmark trained on large-scale datasets for generalizable models to boost their capabilities and conduct cross-subject validation; 4) we analyze the key elements of either animatability or generalizability and propose the first baseline for animatable and generalizable human body rendering from monocular videos. i.e., a generalizable HumanNeRF. We hope our studies and benchmarks could benefit future works.

## 2   Related Work

The neural radiance field, i.e., NeRF [33], is a powerful implicit representation of 3D scenes. There is a series of variants that improve the vanilla NeRF in several aspects, including improving rendering quality [2, 3, 43], reducing train views [35, 23, 42, 58, 53], acceleration [52, 34, 40, 41, 14, 6], one-shot training [46, 7, 30, 22, 9], mesh reconstruction [51, 17, 13, 48, 56, 45, 36] and so on. Among these variants, NeRFs for human body rendering [38, 49, 12, 54, 10, 37, 32, 8, 24] have attracted a lot of attention due to their broad applications. By exploiting the human body prior, they achieve impressive quality for synthesizing high-fidelity human images given sparse view video sequences.

### 2.1   Scene-specific NeRFs for Human

Scene-specific methods [38, 37, 47, 49, 54, 10, 20] for human body rendering require only sparse view videos for training as different video frames can be treated equivalent to dense view images by exploiting the human body prior. The number of train views has been reduced from 100 for

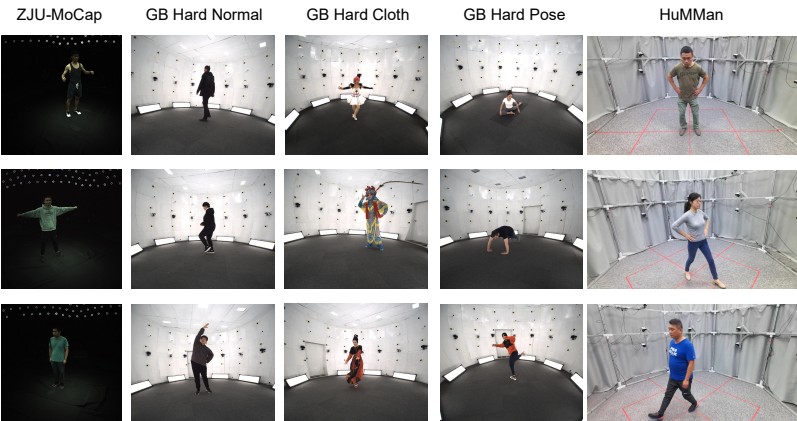

| ZJU-MoCap | GB Hard Normal | GB Hard Cloth | GB Hard Pose | HuMMan |

Figure 1: Example Images of the used datasets ZJU-MoCap, GeneBody, and HuMMan.

vanilla NeRF to 4 for NeuralBody [38] and even 1 for HumanNeRF [49]. The most commonly used human body prior is the SMPL model [31], which parameterizes the human body shape and pose and provides the optimized skinning weights to describe the deformation of the human body. To strengthen the performance of novel pose rendering, some works [37, 54] introduce additional constraints to learn more reasonable skinning weights. UV-Volume[10] proposes a new representation of UV volume, which is used to regress uv coordinates to retrieve color from texture stacks, reduce training time, and support texture editing.

## 2.2 Generalizable NeRFs for Human

Unlike scene-specific methods that take a long time for training until converge, generalizable NeRFs [24, 32, 8, 11, 12, 18, 21] for human body rendering train a model in a one-shot manner. Once a generalizable model completes the training process, it can be directly used to render an unseen human by giving input images of the human in a feed-forward way. Thanks to SMPL vertices that approximate the human body surface, these generalizable methods can project SMPL vertices to image planes to retrieve image features for each vertice. The positions of these vertices are strong prior to geometry inference, making generalization available. NHP [24] projects vertices not only to other image views but also to other nearby frames to get spatial-temporal features. GP-NeRF [8] learns embeddings anchored on vertices to guide the feature fusion from different views. MPSNeRF [16] additionally defines a canonical space and first warp retrieved image features to the space and then to the target pose space so as to achieve animatability. These methods require multi-view images as input, while some works [12, 18] also have explored building a generalizable NeRF model that takes a single image as input. However, there is still no generalizable and animatable method that takes monocular video frames as input. More importantly, existing methods have various settings, making fair comparisons hard.

## 3 Benchmarking Neural Human Radiance Fields

### 3.1 Unifying Evaluation Metrics

The commonly used metrics to measure the difference between a rendered image and a GT image are peak signal-to-noise ratio (PSNR ↑), structural similarity index (SSIM ↑), and learned perceptual image patch similarity (LPIPS ↓) [55]. PSNR measures pixel-wise similarity between the rendered image and the GT image. In contrast, SSIM and LPIPS estimate the patch-wise error between two images. Recent research found that LPIPS is more consistent with human perception. However, some existing works did not report the metric. We add this metric for all experiments to make the quantitative results more credible.

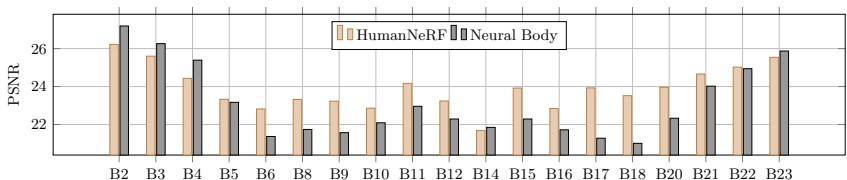

Figure 2: The PSNR distribution among different test views. We use the images from Camera B1 for training, and the rest cameras are located uniformly in a round from B1 to B23.

## 3.2 Evaluating on Challenging Datasets

The most commonly used dataset for neural human body rendering is ZJU-MoCap [38], which consists of 10 multi-view video sequences captured by 24 rounding synchronized cameras. However, most of the subjects perform simple actions and wear close-fitting clothes, and the light condition is also biased to dark and blue effect, as shown in Fig. 1. Moreover, the number of human actors is too few to train a generalizable method. To build more challenging and representative benchmarks, we select the GeneBody [11] and HuMMan [4] datasets. GeneBody contains human actors doing hard poses and wearing complicated clothes (e.g., long dresses), which is suitable for evaluating the real performance of scene-specific methods. HuMMan is a large-scale (about 153 released subjects) dataset with multi-view human video sequences that will benefit generalizable models.

Table 2: Quantitative comparison of representative scene-specific methods with different training view settings on ZJU-MoCap. In this experiment, the number of train frames is fixed to be 300.

| Methods | Single View | | | Multiple views | | |
| --- | --- | --- | --- | --- | --- | --- |
| | PSNR ↑ | SSIM ↑ | LPIPS ↓ | PSNR ↑ | SSIM ↑ | LPIPS ↓ |
| NeuralBody [38] | 22.65 | 0.8667 | 0.1707 | **27.82** | **0.9396** | 0.1048 |
| AniNeRF [37] | 21.48 | 0.8477 | 0.2229 | 24.13 | 0.8822 | 0.1867 |
| HumanNeRF [49] | **23.08** | **0.8763** | **0.1326** | 25.86 | 0.9245 | **0.0728** |

## 3.3 Unifying Evaluation Settings

According to various settings shown in Tab. 1, we make them unified for clear comparisons.

**Normalized metrics via human cropping.** Some methods evaluate the three metrics mentioned above on the entire image [49] while others only calculate on the local human area via human cropping [38]. To avoid small-sized humans in an image and make the comparison unified, we use the cropping setting as normalized metrics in all experiments.

**Train & test view splits.** The default number of train views for scene-specific methods is different due to their different targets (e.g., for monocular videos). To study the effect of the number of train views, we set the number of train views to be one and four to represent the single-view setting and multi-view setting respectively. In Tab. 2, we compare the results of novel view rendering on the ZJU-MoCap dataset. In the single-view training setting, HumanNeRF obtains the best results in all three metrics, especially in the LPIPS metric. When multiple views (i.e. four views) are available for training, NeuralBody performs best on PSNR (↑) and SSIM (↑) while HumanNeRF still has the lowest LPIPS (↓). Since monocular videos are easier to obtain, the single-view setting is more friendly for real applications. However, training only on one view would inevitably lead to bad performance in test views that has large position and angle deviation with the train view. We compute PSNR for each test view in Fig. 2 and find that the PSNR of the rendered images worsens when the test view is far from the trained views. Thus, how to handle spatial-temporal video information to obtain better quality is key for future work.

**Train & test frame splits.** For scene-specific methods, multi-view video frames are divided into two parts: training and novel pose rendering testing. For instance, a multi-view human video sequence may contain 500 frames captured by 24 cameras. The first 300 frames captured by four cameras are

Table 3: Quantitative comparison of representative scene-specific methods with different numbers of train frames on ZJU-MoCap. In this experiment, the number of train views is fixed to be 4.

| Methods | 1 frame | | | 60 frames | | | 100 frames | | | 300 frames | | | 500 frames | | |
|---|---|---|---|---|---|---|---|---|---|---|---|---|---|---|---|
| | PSNR ↑ | SSIM ↑ | LPIPS ↓ | PSNR ↑ | SSIM ↑ | LPIPS ↓ | PSNR ↑ | SSIM ↑ | LPIPS ↓ | PSNR ↑ | SSIM ↑ | LPIPS ↓ | PSNR ↑ | SSIM ↑ | LPIPS ↓ |
| NeuralBody [38] | 23.83 | 0.8881 | 0.1412 | 26.29 | **0.9248** | 0.1087 | **27.35** | **0.9364** | 0.0974 | **27.82** | **0.9396** | 0.1048 | **27.77** | **0.9380** | 0.1137 |
| AniNeRF [37] | 24.56 | 0.8987 | 0.1298 | 24.45 | 0.8899 | 0.1652 | 24.78 | 0.8959 | 0.1558 | 24.13 | 0.8822 | 0.1867 | 24.39 | 0.8865 | 0.1882 |
| HumanNeRF [49] | **25.98** | **0.9031** | **0.1015** | **26.49** | 0.9225 | **0.0800** | 26.40 | 0.9253 | **0.0730** | 25.86 | 0.9246 | **0.0728** | 25.52 | 0.9218 | **0.0754** |

used to train the model. In comparison, the first 300 frames captured by the remaining 20 cameras are used for testing novel view rendering, and the left 200 frames captured by all 24 cameras are used for testing novel pose rendering. With different numbers of train frames, the performance for scene-specific methods is shown in Tab. 3. Given any frames, HumanNeRF can obtain the lowest LPIPS with visually better renderings. For AniNeRF, there is no trend that more frames can get better performance. Notably, the quantitative results of these methods vary in a non-negligible range. Thus, unifying the train and test frames is essential.

After unifying the human cropping, train & test view (4 vs 20), and frame split (300 train frames vs 200 novel pose frames), we compare the quantitative results of recent human body rendering methods (some methods like [21, 29] requiring additional processed data are not included in this table) reported in their original paper with the results we retrained in unified settings on ZJU-MoCap in Tab. 4. Besides the evaluation settings, we also unify the train and test subjects split for generalizable methods, and the quantitative results are obtained on the test subjects split (subject 387, 393 and 394 of the ZJU-MoCap dataset).

Table 4: Unified evaluation of free-view rendering for existing methods on ZJU-MoCap dataset. We compare the qualitative results reported in their original paper and the results after we unify the human cropping, train & test view (4 vs 20), and frame split (300 train frames vs 200 novel pose frames). And for all methods, we additionally calculate the metric of LPIPS, which is more consistent with human eyes.

| | Methods | Reported in paper | | | Unified evaluation | | |
|---|---|---|---|---|---|---|---|
| | | PSNR ↑ | SSIM ↑ | LPIPS ↓ | PSNR ↑ | SSIM ↑ | LPIPS ↓ |
| Scene-specific | NeuralBody [38] | 28.10 | 0.9440 | - | **27.80** | **0.9400** | 0.1050 |
| | AniNeRF [37] | 27.10 | 0.9490 | - | 24.10 | 0.8820 | 0.1870 |
| | HumanNeRF [49] | **30.20** | **0.9680** | 0.0317 | 25.86 | 0.9245 | 0.0728 |
| | UV-Volume [10] | 27.95 | 0.9346 | 0.0720 | 26.45 | 0.9282 | **0.0726** |
| Generalizable | NHP [24] | 24.80 | 0.9050 | - | 24.68 | 0.9034 | 0.1706 |
| | GP-NeRF [8] | **26.00** | **0.9210** | - | **26.66** | **0.9263** | 0.1256 |
| | KeypointNeRF [32] | 25.03 | 0.8969 | - | 26.01 | 0.9159 | **0.1041** |

## 3.4 Analyzing the Effects of Estimated Mask & SMPL

Most existing human body rendering methods depend on using ground-truth human masks and SMPL parameters provided by datasets. For example, NeuralBody exploits SMPL vertices to define latent codes and perform sparse convolution with the coordinates of these vertices as input. GP-NeRF [8] projects posed SMPL vertices on image planes to retrieve image features. Human masks for each image are used to eliminate the impact of background pixels. In these methods, the masks and SMPL parameters are assumed to be given and precise. In practical scenarios, we should estimate masks and SMPL parameters via existing state-of-the-art models. Nevertheless, none of the methods quantitatively investigated the effect of inaccurate masks and SMPL parameters.

To study this issue, we adopt the recent state-of-the-art segmentation method RobustVideoMatting [28] to acquire human masks and human shape estimation method Hybrik [25] to obtain SMPL parameters. To simplify the cases, we conduct experiments on the subjects of ZJU-MoCap. Tab. 5 compares the performance between GT masks and estimated masks. All methods have a slight performance drop with estimated human masks, indicating existing methods have a good tolerance for the estimated mask. Interestingly, the degradation is smaller for multi-view models than single-view models since these methods can correct human masks automatically with aggregated multi-view information.

In comparison, methods suffer from a significant performance drop with the estimated SMPL parameters inputs from Tab. 6, especially for NeuralBody and GP-NeRF. HumanNeRF is more

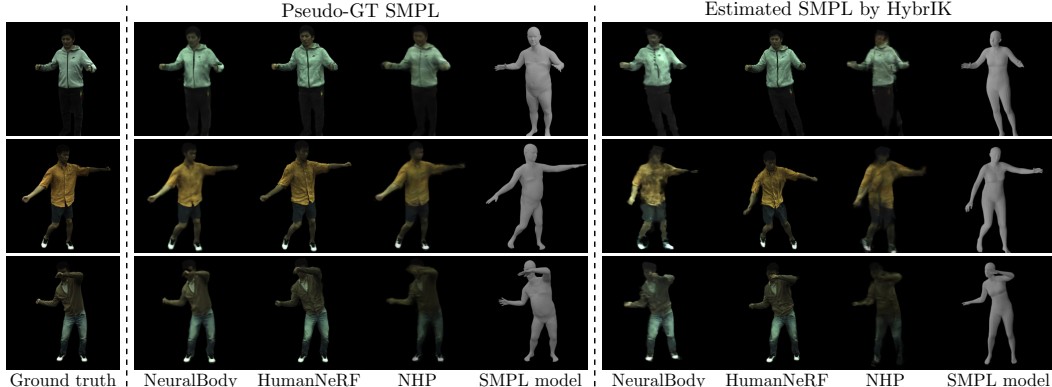

Figure 3: Qualitative comparison of NueralBody, HumanNeRF, and NHP with accurate (pseudo-GT SMPL) and inaccurate (estimated by Hybrik) SMPL parameters.

robust than other methods. Their model designs can explain the phenomenon. NeuralBody defines latent codes on each SMPL vertice, which implicitly stores the appearance information of the human actor. So, inaccurate SMPL will result in heavy changes in learning latent codes during training. In contrast, HumanNeRF stores appearance information of a human implicitly in canonical space, which is irrelevant to SMPL parameters, and its body deformation mechanism is modeled by two parts: rigid transformation, which relies on SMPL parameters, and non-rigid transformation, which is learned implicitly. Therefore, for HumanNeRF, inaccurate SMPL parameters only increase the difficulty of learning the non-rigid transformation but will not severely hurt the human appearance modeling. We also present the qualitative comparisons of NueralBody and HumanNeRF with accurate and inaccurate SMPL parameters in Fig. 3. The texture and pose quality will degrade significantly for NeuralBody and NHP, while the pose quality mainly influences HumanNeRF if the input SMPL parameters suffer high errors.

Table 5: Quantitative comparison of methods using GT masks and using estimated masks to test the tolerance of representative methods to inaccurate masks. We conduct experiments on three subjects of ZJU-MoCap and use RobustVideoMatting as human segmentation.

|  | Methods | GT Mask | | | Estimated by RobustVideoMatting | | |
|---|---|---|---|---|---|---|---|
|  |  | PSNR ↑ | SSIM ↑ | LPIPS ↓ | PSNR ↑ | SSIM ↑ | LPIPS ↓ |
| Scene-specific | NeuralBody [38] | **27.24** | **0.9320** | 0.1205 | **26.93** | **0.9268** | 0.1272 |
|  | HumanNeRF [49] | 22.96 | 0.8793 | **0.1131** | 22.35 | 0.8724 | **0.1257** |
| Generalizable | NHP [24] | 24.68 | 0.9034 | 0.1706 | 24.55 | 0.9005 | 0.1744 |
|  | GP-NeRF [8] | **26.66** | **0.9263** | **0.1256** | **26.33** | **0.9241** | **0.1266** |

Table 6: Quantitative comparison of methods using Pseudo-GT SMPL parameters and using estimated SMPL parameters to test the tolerance of representative methods to inaccurate SMPL parameters. We conduct experiments on three subjects of ZJU-MoCap and use Hybrik as SMPL paramteter estimator.

|  | Methods | Pseudo-GT SMPL (Provided by dataset) | | | Estimated by HybrIK | | |
|---|---|---|---|---|---|---|---|
|  |  | PSNR ↑ | SSIM ↑ | LPIPS ↓ | PSNR ↑ | SSIM ↑ | LPIPS ↓ |
| Scene-specific | NeuralBody [38] | 22.41 | 0.8688 | 0.1583 | 16.65 | 0.7645 | 0.3338 |
|  | HumanNeRF [49] | **22.96** | **0.8793** | **0.1131** | **21.87** | **0.8697** | **0.1272** |
| Generalizable | NHP [24] | 24.68 | 0.9034 | 0.1706 | **19.47** | 0.7914 | 0.3170 |
|  | GP-NeRF [8] | **26.66** | **0.9263** | **0.1256** | 19.02 | **0.8045** | **0.3093** |

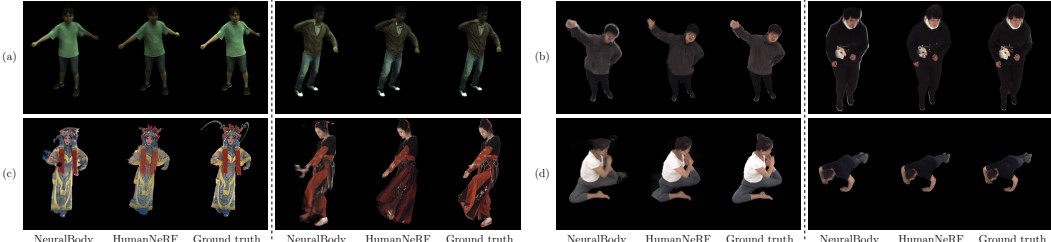

| NeuralBody | HumanNeRF | Ground truth | NeuralBody | HumanNeRF | Ground truth | NeuralBody | HumanNeRF | Ground truth | NeuralBody | HumanNeRF | Ground truth |

Figure 4: Qualitative comparisons on the widely used ZJU-MoCap (a) and challenging GeneBody datasets (b,c,d) on normal, hard clothes, and hard poses scenes.

# 4    Analyses of Generalization and Animatability

This section empirically investigates the generalization ability and animatability of existing human body rendering methods. For scene-specific methods, we select NeuralBody and HumanNeRF as representative methods; for generalzable methods, we select GP-NeRF as the representative method.

## 4.1    Benchmarking Scene-specific Methods on GeneBody

For scene-specific methods of human body rendering, the meaning of generalization is two folds: novel view rendering and novel pose rendering. Novel view rendering renders novel views of a human in a pose seen during training. In comparison, novel pose rendering needs to render a human in a given unseen pose from novel views. In general, novel pose rendering is more challenging than novel view rendering because rendering a human in a novel pose requires a method to model the human body deformation well. Unfortunately, few scene-specific methods have sufficient experiments and discussions for the two aspects separately. Moreover, human actors in ZJU-MoCap only wear easy close-fitting clothes and perform simple actions with small-scale subjects, so the dataset is hard to reflect the real generalization ability. Therefore, as discussed above, we adopt the GeneBody dataset and conduct extensive experiments to explore the effectiveness of representative scene-specific methods. We evaluate the performance of novel view and pose rendering separately in Tab. 7. Following the official splits, we also split the dataset into normal, hard cloth, and hard pose. Meanwhile, we unify the number of train views to be four and train frames to be 100 while the remaining views and frames are used for evaluation. There are some observations from the results:

*1. The tolerance to SMPL inaccuracy varies among different methods.* In the GeneBody dataset, the provided masks and SMPL parameters are not as accurate as ZJU-MoCap. So, we can see from the columns of **GeneBody Normal** even though the human actors do not wear complex clothes or perform hard poses, NeuralBody still performs poorly due to inaccurate SMPL parameters. The reason behind this has been discussed in Sec. 3.4.

*2. Existing scene-specific methods still perform poorly in challenging cases.* As demonstrated in the columns **GeneBody Hard Cloth** and **GeneBody Hard Pose**, both NeuralBody and HumanNeRF have a severe performance drop on all metrics compared to the results on ZJU-MoCap. The challenges brought by hard clothes are two folds. The first is that the motion of clothes (especially loose clothes) cannot be depicted by SMPL model. The other one is that complex clothes have more texture details and geometry variations. They lead to increased difficulties for both appearance and body deformation modeling in the sense of more data inconsistency. As for hard poses, the challenge is the larger variations of body and the corresponding cloth deformation, which increases the difficulties for body deformation modeling.

*3. Compared to hard poses, hard clothes are more challenging for existing models.* Comparing the column **GeneBody Hard Cloth** and **GeneBody Hard Pose**, it can be observed that hard clothes have a more severe impact on the overall performance for all representative methods than hard poses. This is because hard clothes affect both appearance and body deformation modeling, while hard poses mainly affect body deformation modeling. The qualitative results are shown in Fig. 4

*4. Existing scene-specific methods have a consistent performance drop for novel pose rendering.* Comparing the results in the top rows (novel view rendering) and bottom rows (novel pose rendering), we can find that with the same settings and training data, the selected representative methods have

inferior performance for novel pose rendering compared to novel view rendering. This indicates that the learned body deformation cannot generalize well enough to unseen poses for existing scene-specific methods, especially significantly increasing the LPIPS.

Table 7: Extensive quantitative comparison of representative scene-specific methods on ZJU-MoCap and different partitions of GeneBody.

| | Novel view rendering | | | | | | | | | | | |
| | ZJU-MoCap | | | GeneBody normal | | | GeneBody hard cloth | | | GeneBody hard pose | | |
| Methods | PSNR ↑ | SSIM ↑ | LPIPS ↓ | PSNR ↑ | SSIM ↑ | LPIPS ↓ | PSNR ↑ | SSIM ↑ | LPIPS ↓ | PSNR ↑ | SSIM ↑ | LPIPS ↓ |
|---|---|---|---|---|---|---|---|---|---|---|---|---|
| NeuralBody [38] | **27.82** | **0.9396** | 0.1048 | 19.31 | 0.8499 | 0.3197 | 16.24 | **0.7748** | 0.3093 | 19.99 | **0.8412** | 0.2467 |
| HumanNeRF [49] | 25.86 | 0.9250 | **0.0728** | **24.63** | **0.8865** | **0.2319** | **17.36** | 0.7729 | **0.2155** | **21.56** | 0.8388 | **0.1715** |
| | Novel pose rendering | | | | | | | | | | | |
| | ZJU-MoCap | | | GeneBody normal | | | GeneBody hard cloth | | | GeneBody hard pose | | |
| Methods | PSNR ↑ | SSIM ↑ | LPIPS ↓ | PSNR ↑ | SSIM ↑ | LPIPS ↓ | PSNR ↑ | SSIM ↑ | LPIPS ↓ | PSNR ↑ | SSIM ↑ | LPIPS ↓ |
| NeuralBody [38] | **23.73** | 0.8871 | 0.1525 | 18.79 | 0.8325 | 0.3402 | 14.47 | 0.7039 | 0.3657 | 17.21 | 0.7801 | 0.2985 |
| HumanNeRF [49] | 23.64 | **0.8907** | **0.1027** | **24.61** | **0.8861** | **0.2254** | **15.60** | **0.7331** | **0.2610** | **19.07** | **0.7946** | **0.2122** |

## 4.2 Benchmarking Generalizable Methods on HuMMan

For generalizable methods, the generalization mainly refers to novel view rendering for unseen human actors after pre-training on the multi-view images of a certain number of human actors. The scale and diversity of training data are of great importance to achieve good generalization. To study the potential of this stream of methods, We evaluate the in-domain (i.e., cross-subject) and out-of-domain (i.e., cross-dataset) generalization ability the selected representative generalizable method, i.e. GP-NeRF.

From Tab. 8, the test results consistently perform better than the model trained on the in-domain dataset. Even using a large-scale HuMMan dataset to train the model and cross-dataset test, the performance is still worse than the model trained on ZJU-MoCap. This indicates that the existing generalizable methods still cannot achieve satisfying performance when rendering novel views for an unseen human in real scenarios. The state-of-the-art generalizable method GP-NeRF tends to overfit small-scale data due to the higher performance on in-domain ZJU-MoCap training and testing.

Table 8: Quantitative comparison of cross-subjects generalization of GP-NeRF. ZJU-MoCap 7 and ZJU-MoCap 3 mean the train and test split of ZJU-MoCap, respectively.

| Train set | Test set | PSNR ↑ | SSIM ↑ | LPIPS ↓ |
|---|---|---|---|---|
| ZJU-MoCap 7 | ZJU-MoCap 3 | **26.66** | **0.9263** | **0.1256** |
| HuMMan | ZJU-MoCap 3 | 23.82 | 0.8810 | 0.1850 |
| ZJU-MoCap 7 | HuMMan Eval | 17.89 | 0.8833 | 0.2318 |
| HuMMan | HuMMan Eval | **21.68** | **0.9234** | **0.1511** |

**Upper-bound performance of finetuning on a single subject.** For generalizable methods, to further increase the performance of novel view rendering for an unseen human, one effective way is to finetune the trained model on the human's multi-view sequences. To test the upper-bound performance of finetuning, we use the pretrained model of GP-NeRF to train three subjects of ZJU-MoCap separately for a long enough time and record the quantitative performance after 15 minutes, 1 hour, and 20 hours, respectively. As shown in Fig. 5 and Tab.9, benefiting from pretraining, GP-NeRF has higher PSNR than HumanNeRF from the early start of finetune but increases slowly even after a long training time. In contrast, the scene-specific method HumanNeRF is trained from scratch with the same settings, and it performs poorly initially, but its PSNR grows faster than GP-NeRF. After enough time for finetune (training), HumanNeRF has significantly lower LPIPS so that the rendered images by HumanNeRF look realistic and high-quality to the GT images.

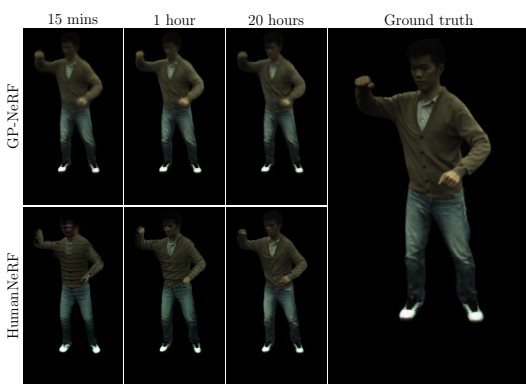

Figure 5: Rendering quality grows with the increase of finetuning time. The generalizable method GP-NeRF initially looks better than the scene-specific method HumanNeRF, but HumanNeRF has visually better results after several hours.

Table 9: Quantitative comparison of the upper bound performance of finetuning on a single subject using a pretrained model for the generalizable method GP-NeRF. For reference, we also train HumanNeRF from scratch given the same train views, frames, and time.

| Methods | 15 mins finetune | | | 1 hour finetune | | | 20 hours finetune | | |
|---|---|---|---|---|---|---|---|---|---|
| | PSNR ↑ | SSIM ↑ | LPIPS ↓ | PSNR ↑ | SSIM ↑ | LPIPS ↓ | PSNR ↑ | SSIM ↑ | LPIPS ↓ |
| GP-NeRF [8] | **27.46** | **0.9275** | **0.1464** | **27.94** | **0.9330** | 0.1390 | **28.80** | **0.9408** | 0.1290 |
| HumanNeRF [49] | 22.22 | 0.8549 | 0.1772 | 23.26 | 0.8779 | **0.1285** | 25.36 | 0.9156 | **0.0807** |

## 4.3 Exploring the Key to Animatablity

**Scene-specific methods.**    For scene-specific methods, the definition of animatability is equivalent to novel pose rendering, i.e., rendering a human in a given pose. Thus, as discussed in Sec.4.1, existing scene-specific methods are all animatable, but the rendering quality would be worse than novel view rendering. The key element of animatability for scene-specific methods is a module to model the body deformation so that they can use images of a human performing different actions for training. For example, NeuralBody encodes the appearance information of a human into SMPL vertices, so it can directly use the body deformation module of SMPL to animate a human body into any given pose. In contrast, HumanNeRF encodes appearance information of a human of T-pose in canonical space and decouples the body deformation into skeletal motion, which is modeled by an implicitly learned LBS (linear blend skinning) weight field, and non-rigid motion, which is modeled by an implicitly learned translation field. Most of the existing scene-specific methods follow these two paradigms. From our experiments in Sec. 3.4, the latter has more tolerance to inaccurate SMPL parameters, while the former is easier to implement.

**GeneHumanNeRF: making HumanNeRF generalizable.**    Existing generalizable methods are not animatable, except for MPSNeRF [16], because they require a certain number of views as input images to render novel views. For generalizable methods, the appearance information of a human to be rendered is acquired from input images. Thus, these methods cannot directly work by merely specifying a pose without reference to multi-view input images.

Table 10: The performance of our built simple baseline GeneHumanNeRF for animatable and generalizable human body rendering from a monocular video.

| Train set | Test set | PSNR ↑ | SSIM ↑ | LPIPS ↓ |
|---|---|---|---|---|
| ZJU-MoCap 7 | ZJU-MoCap 3 | **22.87** | **0.8647** | **0.2110** |
| HuMMan | ZJU-MoCap 3 | 22.73 | 0.8573 | 0.2290 |
| ZJU-MoCap 7 | HuMMan Eval | 18.09 | 0.8715 | 0.2159 |
| HuMMan | HuMMan Eval | **21.06** | **0.8882** | **0.2010** |

MPSNeRF achieves animatability by exploiting LBS warping twice so that the appearance information of a posed human in the input multi-view images can be warped to the rendered human in given poses. The key element is a canonical pose space where the appearance information from input images is first warped to and from which the appearance information is warped to the target pose space. We find that such an element is not limited to the setting that requires multi-view images as input. Therefore, borrowing this effective module, we built a new generalizable and animatable method that takes a few monocular video frames as input (e.g., [16]), which is like an extension of HumanNeRF from the setting of per-scene trained into one-shot trained. We name this new baseline method as GeneHumanNeRF. The motivation for this baseline method is that monocular videos are easier to acquire, so it is more suitable for practical applications. The more technical details of this baseline can be found in supplementary material. We build the benchmark on both ZJU-MoCap and HuMMan datasets and evaluate their cross-subject and cross-dataset generalization ability in Tab. 10. The performance of the proposed simple baseline can achieve a competitive performance on both datasets, which can be a good reference for future works.

# 5   Inspirations for Future Works

## 5.1   Unifying Settings

As we can see from the tables presented in Sec. 3, chaotic train & evaluation settings make the community hard to distinguish the advantages and drawbacks of existing methods from their paper-reported results. And our ablation studies for different settings indicate that those settings can make a difference. Therefore, for future works, it is highly recommended to conduct experiments in the

settings consistent with those used in this benchmark and report their results of more settings (more datasets, different frame & view settings and so on). In addition, most works assume accurate SMPL parameters are available by default, but it is not true especially for real applications. Thus, how to reduce the adverse impact of inaccurate SMPL parameters can also be a problem worthy of studying.

## 5.2 Scene-specific Methods

For scene-specific methods, future works should focus more on the scenes that people wearing complex clothes and improve the pose generalization. To evaluate the real performance on these aspects, more datasets should be included for experiments. Besides, a proper sampling strategy for train frames and views can also lead to a significant performance increase. As most existing methods rely on SMPL to capture the human motions, it is worth thinking is there any alternative model that can depict the human geometry more precisely and whose body parts move more naturally.

## 5.3 Generalizable Methods

For generalizable methods, future works should focus on the improvement of cross-dataset generalization. Currently, existing methods train on small-size datasets and evaluate on the in-domain test sets. Although the quantitative results look good, they have poor performance when tested on other datasets that have more diverse cases. Due to the high cost for acquiring multi-view video sequences, data scarcity may also be an obstacle to training a generalizable model that can perform well in real scenarios. On the other hand, animatability is not compatible with the common settings of existing generalizable methods, but animatability is expected for many applications. Therefore, building a high-performance animatable and generalizable model that takes a monocular video as input can lead to interesting applications. We have proposed a simple baseline of this setting to be a reference for future works.

# 6 Conclusion

In this work, we build the inaugural comprehensive benchmark for neural human radiance fields. Specifically, we unify the evaluation settings and metrics. We introduce more challenging datasets and establish the benchmark for them. To explore the capability of existing generalizable models, we train them on large-scale datasets and conduct cross-subject validation. Lastly, after analyzing the key components of animatability or generalizability, we design a baseline that caters to both these attributes given monocular videos. We sincerely hope these efforts can benefit this field of study.

**Acknowledgement**   The work was supported in part by NSFC with Grant No. 62293482, the Basic Research Project No. HZQB-KCZYZ-2021067 of Hetao ShenzhenHK S&T Cooperation Zone, the National Key R&D Program of China with grant No. 2018YFB1800800, by Shenzhen Outstanding Talents Training Fund 202002, by Guangdong Research Projects No. 2017ZT07X152 and No. 2019CX01X104, by the Guangdong Provincial Key Laboratory of Future Networks of Intelligence (Grant No. 2022B1212010001), and by Shenzhen Key Laboratory of Big Data and Artificial Intelligence (Grant No. ZDSYS201707251409055). It was also partially supported by NSFC62172348, Outstanding Young Fund of Guangdong Province with No. 2023B1515020055 and Shenzhen General Project with No. JCYJ20220530143604010. In addition, we really thank Yuqi Hu from HKUST (GZ) for helping code cleaning and build the project webpage.

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
