# —Supplementary Material—
# A Comprehensive Benchmark for Neural Human Radiance Fields

**Kenkun Liu**[1,2*]**, Derong Jin**[2]**, Ailing Zeng**[1†]**, Xiaoguang Han**[2]**, Lei Zhang**[1]

[1]International Digital Economy Academy (IDEA)
[2]The Chinese University of Hong Kong, Shenzhen
`https://kenkunliu.github.io/HMNeRFBench/`

Due to space constraints in the main paper, We elaborate on the following in the supplementary material: (1) more ablation study of different settings in Sec. 1; (2) more results of scene-specific methods on HuMMan in Sec. 2; (3) more results of generalizable methods on GeneBody in Sec. 3; (4) experimental details in Sec. 4; (5) details in our proposed generalizable and animatable model *GeneHumanNeRF* in Sec. 5; (6) discussions on limitations in Sec. 6; and (7) potential negative social impact in Sec. 7.

## 1 More Ablation Study of Different Settings

**Train frame sampling intervals.** For multi-view video sequences, consecutive frames may have severe information redundancy that is adverse to the reconstruction of the human geometry and appearance. Thus, we fix the number of train frames and train views but set the frame sampling interval to be 1, 5 and 10. The experimental results are shown in Tab. 1. In general, with larger sampling interval, the performance of selected representative scene-specific methods would increase.

Table 1: Quantitative comparison of representative scene-specific methods with different numbers of train frame sampling intervals on ZJU-MoCap. In this experiment, the number of train views is fixed to be 4 and the number of train frames to be 60.

| Methods | interval 1 | | | interval 5 | | | interval 10 | | |
|---|---|---|---|---|---|---|---|---|---|
| | PSNR ↑ | SSIM ↑ | LPIPS ↓ | PSNR ↑ | SSIM ↑ | LPIPS ↓ | PSNR ↑ | SSIM ↑ | LPIPS ↓ |
| NeuralBody [5] | 26.29 | **0.9248** | 0.1087 | **27.75** | **0.9394** | 0.0947 | **27.78** | **0.9393** | 0.0967 |
| AniNeRF [4] | 24.45 | 0.8899 | 0.1652 | 24.54 | 0.8872 | 0.1746 | 25.12 | 0.8991 | 0.1695 |
| HumanNeRF [6] | **26.49** | 0.9225 | **0.0800** | 25.84 | 0.9251 | **0.0716** | 25.75 | 0.9249 | **0.0745** |

**Train view distributions.** We visualize the camera locations and orientations of ZJU-MoCap dataset in Fig. 1, where the cameras of train views are uniformly located (the red cameras are selected as train views while the remaining are as test views). We also conduct the experiments when the 4 cameras of train views are located in half-uniform, quarter-uniform and neighbouring places, the results are shown in Tab. 2. When the selected train views are too close to each other, the performance of NeuralBody and AniNeRF drops severely.

**Train time.** In this part, we fix other settings and train the representative scene-specific methods with different lengths of time to compare the convergence speed of each method. The results are shown in Tab. 3. We can see that NeuralBody and AniNeRF achieve good performance within 1 hour but cannot benefit from longer train time while HumanNeRF's performance rises with the increase of train time, which indicates HumanNeRF requires much more time to converge.

---

*Work done during an internship at IDEA.
†Corresponding author.

Uniform distribution of cameras in 3D Space

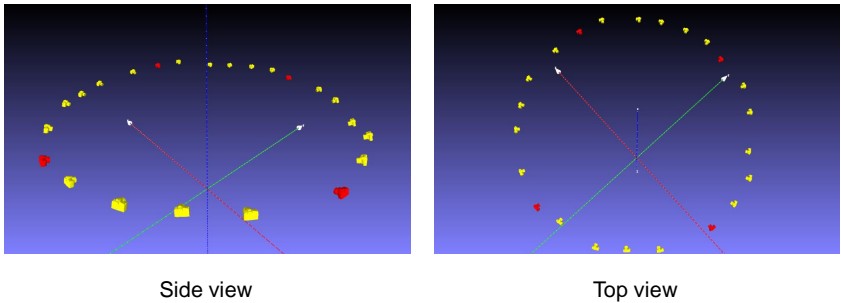

Side view              Top view

Figure 1: The visualization of camera positions and orientations of ZJU-MoCap dataset in 3D space.

Table 2: Quantitative comparison of representative scene-specific methods with different train view distribution on ZJU-MoCap. In this experiment, the number of train views is fixed to be 4 and the number of train frames to be 300.

| Methods | uniform | | | half-uniform | | | quarter-uniform | | | neighbouring | | |
| | PSNR ↑ | SSIM ↑ | LPIPS ↓ | PSNR ↑ | SSIM ↑ | LPIPS ↓ | PSNR ↑ | SSIM ↑ | LPIPS ↓ | PSNR ↑ | SSIM ↑ | LPIPS ↓ |
|---|---|---|---|---|---|---|---|---|---|---|---|---|
| NeuralBody [5] | **27.82** | **0.9395** | 0.1048 | **27.18** | **0.9339** | 0.1084 | **25.96** | **0.9190** | 0.1213 | 21.72 | 0.8590 | 0.1823 |
| AniNeRF [4] | 24.12 | 0.8822 | 0.1867 | 23.82 | 0.8792 | 0.1902 | 23.09 | 0.8687 | 0.2010 | 21.24 | 0.8416 | 0.2272 |
| HumanNeRF [6] | 25.86 | 0.9246 | **0.0727** | 25.74 | 0.9227 | **0.0749** | 25.09 | 0.9142 | **0.0869** | **24.03** | **0.8958** | **0.1116** |

## 2 More Results of Scene-specific Methods on HuMMan

We also conduct experiments for scene-specific methods on the part of HuMMan dataset. From the dataset, we select a certain number of male and female multi-view video sequences and measure the performance of the representative methods for both novel view rendering and novel pose rendering. The quantitative results are shown in Tab. 4

## 3 More Results of Generalizable Methods on GeneBody

In Tab. 5, we present more results to evaluate the cross-subjects generalization for the selected representative generalizable method GP-NeRF, from which we can see the consistent trend with scene-specific methods on the GeneBody dataset.

## 4 Experiment Details

We use a single RTX 3090 Ti GPU to run all our experiments. For scene-specific methods, the training time for a single scene is within 24 hours. For generalizable methods, the training time for pretraining is within 48 hours. For unified evaluation of scene-specific methods, we set the number of train views and train frames to be 4 and 300, respectively, on ZJU-MoCap, and 4 and 100, respectively, on GeneBody, and 4 and around 50, respectively, on HuMMan. For generalizable methods, we follow the officially recommended train and test split for HuMMan, and the split used in NHP [3] for ZJU-MoCap.

## 5 Details of GeneHumanNeRF

The overview of our built GeneHumanNeRF is shown in Fig. 2. The input of our model is a sequence of input frames containing a moving human. A few layers of CNN are adopted to extract image features, which are used to construct a 3D mesh feature volume in canonical space with the help of the human body prior. Given a user-controlled pose and a camera view, a batch of rays is cast from the camera center passing through the target space, where the spatial points are sampled. Each point will be transformed via LBS warping to canonical space to retrieve canonical features and to the observation spaces to retrieve pixel-aligned information. With the two kinds of information, the color

Table 3: Quantitative comparison of representative scene-specific methods with different train time on ZJU-MoCap. In this experiment, the number of train views is fixed to be 4 and the number of train frames to be 300.

| Methods | 1 hour PSNR ↑ | SSIM ↑ | LPIPS ↓ | 5 hours PSNR ↑ | SSIM ↑ | LPIPS ↓ | 10 hours PSNR ↑ | SSIM ↑ | LPIPS ↓ | 20 hours PSNR ↑ | SSIM ↑ | LPIPS ↓ |
|---|---|---|---|---|---|---|---|---|---|---|---|---|
| NeuralBody [5] | **27.70** | **0.9374** | 0.1180 | **27.88** | **0.9402** | 0.0998 | **27.86** | **0.9404** | 0.0947 | **27.84** | **0.9403** | 0.0934 |
| AniNeRF [4] | 24.17 | 0.8810 | 0.1909 | 24.06 | 0.8830 | 0.1845 | 23.92 | 0.8837 | 0.1829 | 23.87 | 0.8838 | 0.1825 |
| HumanNeRF [6] | 23.77 | 0.8900 | 0.1191 | 24.97 | 0.9137 | **0.0871** | 25.52 | 0.9210 | **0.0778** | 25.85 | 0.9252 | **0.0729** |

Table 4: Extensive quantitative comparison of representative scene-specific methods on HuMMan [1]. We select 5 male multi-view video sequences and 5 female multi-view video sequences for experiments and average their quantitative results of novel view and pose rendering, respectively. The number of train views is set to four, while the number of train frames is around 50 due to the limited original frames.

| | Novel view rendering | | | | | |
|---|---|---|---|---|---|---|
| | HuMMan Male | | | HuMMan Female | | |
| Methods | PSNR ↑ | SSIM ↑ | LPIPS ↓ | PSNR ↑ | SSIM ↑ | LPIPS ↓ |
| NeuralBody [5] | **22.87** | **0.9324** | 0.1096 | **20.59** | **0.9155** | 0.1341 |
| HumanNeRF [6] | 22.84 | 0.9299 | **0.0826** | 20.46 | 0.9081 | **0.0965** |

| | Novel pose rendering | | | | | |
|---|---|---|---|---|---|---|
| | HuMMan Male | | | HuMMan Female | | |
| Methods | PSNR ↑ | SSIM ↑ | LPIPS ↓ | PSNR ↑ | SSIM ↑ | LPIPS ↓ |
| NeuralBody [5] | 20.45 | 0.9070 | 0.1304 | **19.00** | **0.8885** | 0.1544 |
| HumanNeRF [6] | **20.74** | **0.9101** | **0.0974** | 18.94 | 0.8829 | **0.1175** |

and density for each point are regressed by an MLP. Finally, the color for a pixel in the target image is synthesized by integrating the sampled points' color and density on the corresponding casted ray via volume rendering. Compared to NHP [3] and GPNeRF [2], which are not animatable, our proposed method GeneHumanNeRF aggregates information from the input images in a canonical space first and then warps it to the target pose space instead of directly transferring information from the input images to the target rendered image. Thus, our method achieves animatability as the user can give the target pose, which can be seen as an extension from scene-specific HumanNeRF to generalizable HumanNeRF.

# 6 Limitations

Although this work comprehensively explores the unified experimental settings on diverse models and novel datasets, the limitation of this benchmark is that currently, we conduct experiments on some representative methods since some recent works are not open-source or have quite different codebase architectures. In the future, we will continue to include more methods with more datasets to expand the benchmark.

Table 5: Quantitative comparison of cross-subjects generalization of GP-NeRF. ZJU-MoCap 7 and ZJU-MoCap 3 mean the train and test split of ZJU-MoCap, respectively.

| Train set | Test set | PSNR ↑ | SSIM ↑ | LPIPS ↓ |
|---|---|---|---|---|
| HuMMan | HuMMan Eval | 21.68 | 0.9234 | 0.1511 |
| HuMMan | ZJU-MoCap 3 | 23.82 | 0.8810 | 0.1850 |
| HuMMan | GB Normal | 18.88 | 0.8125 | 0.3995 |
| HuMMan | GB Hard Cloth | 15.06 | 0.7277 | 0.3853 |
| HuMMan | GB Hard Pose | 16.56 | 0.7672 | 0.3391 |

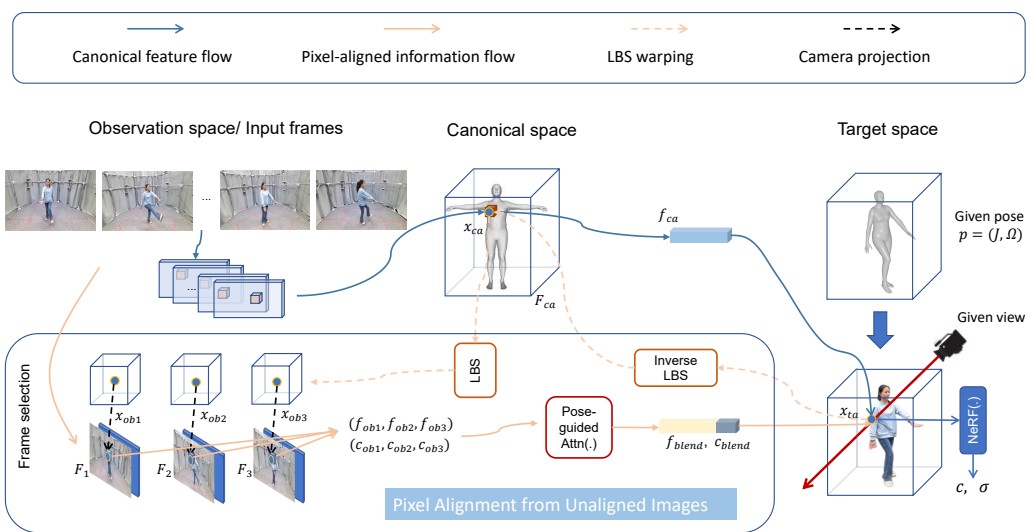

Figure 2: The overview of our proposed GeneHumanNeRF, the first generalizable and animatable method from monocular videos.

# 7 Potential Negative Social Impact

We are studying the task of human body rendering, which may be used to fabricate fake videos that could raise issues around digital identity and privacy. These implications need to be taken into account when developing and deploying these systems.