# OpenReview forum: "A Comprehensive Benchmark for Neural Human Radiance Fields"
_NeurIPS.cc/2023/Track/Datasets_and_Benchmarks — NeurIPS 2023 Datasets and Benchmarks Poster_

### Official Review · Reviewer_1eYJ · 2023-07-10
**Well-structured benchmark for human-oriented NeRFs**

**Rating:** 7
**Confidence:** 4

**Strengths:**

I believe with the rapid growth of novel view rendering field in recent years, this paper addresses an important issue of unifying benchmarks, and providing a degree of comparability between various approaches. There are several factors valuable for the research community which are listed below.

Firstly, Table 4 (comparison between results reported in paper and unified evaluation) highly supports the need for a more unified benchmarking procedure. The authors used probably the currently most common dataset for evaluating NeRF for humans - ZJU-MoCap [1], and proposed a fixed set of training and testing views, and frames used for evaluation, followed by retraining of the human-oriented NeRF models to show how different the results can be in such a setting. Also, the authors noted that some methods provide metrics calculated on full vs cropped version of images, which is also worth clarifying, especially given that human datasets provide images with uniform background and typically results will be much higher for such with respect to their cropped counterparts.

Further strenght of the paper is the analysis of the effects of human mask and SMPL parameters estimation in contrast to the use of ground truth values. This is a very useful analysis for in-the-wild use of the evaluated methods. Specifically, the changes in performance for the estimated vs GT SMPL parameters is interesting for the community as the quality of estimation would be expected to vary significantly with the number of available views, which is an important restriction in real world setups.

Separation of GeneBody [2] dataset is an interesting approach, splitting data based on hard clothing or poses seems to be a very sensible approach. Additionally, the results of novel view and pose rendering (Table 7) support the intuition that loose clothing provide more challenge to the tested models than hard poses due to the nature of the models that include explicit human pose representation within.

With the emergence of generalisable NeRF models, structuring the evaluation protocol is also important (specifically choice of training and testing identities). Following that the experiment with cross-dataset testing is very interesting as it can be a reasonable proxy to the real-world scenario.

Finally, the authors propose a generalisable version of HumanNeRF [3] which allows for animatability from a monocular video. I believe this to be a good baseline for future methods exploring such a setup.

To summarise the strengths, the authors of the paper put a significant effort towards unifying human NeRF benchmarking, including retraining and analysing several current methods. The results of the benchmark and insights show the need of such a benchmark for making the research comparable between each other.

**Additional Feedback:**

Some questions:

Did you observe any correlation between HumanNeRF performance and the range of movements in the training examples. In our experiments, we noticed that with low coverage of movement around body axis (e.g. mostly frontal view) the performance drops drastically.

In Table 9, Figure 4, and mentioned in line 263 HumanNeRF after longer training provides more visually pleasing results. However, we notice significantly smaller PSNR. Do you have any qualitative observation of why that may happen. Is there a color shift between images or human translation not clearly visible with the naked eye?

To summarise, I don't see any major flaws in this work, rather a couple of points for clarification. I would really like the authors to describe how the benchmark will look like after publication as this would be the most useful to research community if being kept updated.

References:

[1] Sida Peng, Yuanqing Zhang, Yinghao Xu, Qianqian Wang, Qing Shuai, Hujun Bao and Xiaowei Zhou. Neural body: Implicit neural representations with structured latent codes for novel view synthesis of dynamic humans. 2021 IEEE/CVF Conference on Computer Vision and Pattern Recognition (CVPR), pages 9050–9059, 2020.

[2] Wei Cheng, Su Xu, Jingtan Piao, Chen Qian, Wayne Wu, Kwan-Yee Lin, and Hongsheng Li. Generalizable neural performer: Learning robust radiance fields for human novel view synthesis. arXiv preprint arXiv:2204.11798, 2022.

[3] Chung-Yi Weng, Brian Curless, Pratul P Srinivasan, Jonathan T Barron, and Ira Kemelmacher-Shlizerman. Humannerf: Free-viewpoint rendering of moving people from monocular video. In Proceedings of the IEEE/CVF Conference on Computer Vision and Pattern Recognition, pages 16210–16220, 2022.

[4] Sida Peng, Yuanqing Zhang, Yinghao Xu, Qianqian Wang, Qing Shuai, Hujun Bao, and Xiaowei Zhou. Neural body: Implicit neural representations with structured latent codes for novel view synthesis of dynamic humans. 2021 IEEE/CVF Conference on Computer Vision and Pattern Recognition (CVPR), pages 9050–9059, 2020.

**Clarity:**

The paper is well-written, it is easy to follow and is well-structured.


**Correctness:**

This is a benchmark paper unifying evaluation protocol between human-oriented NeRF methods. It is constructed in a sound way and tries to address several issues concerning various methods (e.g. impact of SMPL estimation on the performance, impact of loose clothing articles). What can be improved is more detailed description of motivation behind design choices.

**Documentation:**

This is a benchmark paper and authors provide a GitHub repository address that contains code corresponding to the experiments in the paper.

**Ethics:**

No ethical concerns.

**Limitations:**

The authors of the proposed benchmark identified performing experiments on the selections of methods as the current limitation of the work. This is most likely the biggest concern with the benchmark paper. The authors mention recent works being not open-source which is perfectly understandable. To this end, it would be good if authors are able to describe how the benchmark would look like after the publication, i.e. would there be a shared codebase and results for all methods published on a website. How would the support for new models and datasets look like?

The potential negative societal impact is briefly described in the supplementary, and as a benchmark paper that is not proposing a new dataset, I believe this is a sufficient explanation.


**Opportunities For Improvement:**

While the paper has a lot of merits and interesting insights, I believe there are several details that could improve this work.

Firstly, as a benchmarking paper, I would like to see more insight on the choices made while unifying the benchmarking procedures. The authors select certain views of the dataset to constitute training and testing splits. It would be nice to be given an explanation of how those views were chosen. Specifically, whether all testing views have similarly spaced neighbours in training, or could some testing views be seen as *easier* or *harder*. Similarly, for novel pose rendering we have a selection of initial frames in the video (which is a reasonable choice). This could use some analysis on the poses in testing frames, i.e. are the poses in test split close to the ones in training (e.g. a percentage of poses withing a given threshold, calculated as distance in parameter space). Such analyses would be useful to explain how the selection process of creating the benchmark was guided.

Following the aforementioned, a clear distinction of all settings should be put in the work, e.g. as a table in the supplementary material covering specific selections of views, frames, ids for generalisation training for all used datasets. This when referring to the paper would help researchers to quickly and clearly prepare a setup for their experiment. (Another idea would be to provide dataloading scripts for those datasets and experimental settings).

Similarly, some more insights can be shared on the selection of hard clothing and hard poses for GeneBody dataset. I believe an idea for such a separation is good. However, for a benchmarking paper, all design choices should be well motivated and described. Namely, in this case what constitutes a hard clothing, are there any clothings that could be easily misclassified by another person. Similarly for pose, is the choice subjective, or is there a guideline behind selection.

A lot of human/object oriented methods use masked PSNR as a metric. The authors propose to follow the idea from NeuralBody [4] of cropping the image around the mask. I understand this enables the use of the same exact rectangular area for calculating the PSNR, SSIM, and LPIPS. Also, this helps if a method produces floating artefacts around human body. However, for completeness e.g. in supplementary, a masked PSNR could be used.

Following on design choices w.r.t. the selected views, what is intuition behind choosing 1 and 4 views for testing, 1 is understandably monocular capture scenario. Does 4 correspond to balancing the number of training/test views in the dataset? Also in lines 134-136 you mention worsening of PSNR with respect to distance. It would be good to show the performance with respect to position and viewing angle values to get a full overview. It can get us more insight into datasets and their design as distances between cameras may be different for different datasets.

In sec. 4.2 a benchmarking of generalisable methods is described. It is a bit unclear which method is tested there (as well as in Table 8), as only the last line of the paragraph mentions the method (line 241). Also why was only one of 2 mentioned methods tested there?

I believe that in a complete benchmark it would be valuable to add training time as an information. This would allow to observe if there exist trade-off between time and performance between methods, and could be useful for researchers to guide their choice of the models to use. The authors mention training times in supplementary (sec. 4), but it would be worth to add whether those are per original methods suggestions and whether training for longer yields significant performance improvements.

The description of the proposed GeneHumanNeRF is rather brief, and could probably use some more details on the setup (but given it only is proposed as a baseline, and the code is provided, I don't consider that as a big issue).

**Relation To Prior Work:**

The paper is related to relatively new focus in the research community (NeRF representations of 3D scenes). I believe authors correctly identified most impactful human-oriented NeRF models, and motivated why and how are they placed in the proposed evaluation.

**Summary And Contributions:**

This work tackles existing issues with evaluation of Novel View Rendering methods for human subjects. The authors observed that comparisons between various NeRFs for humans are difficult due to the use of different datasets, or even different splits or settings in the same datset. Therefore, this work tries to establish a common benchmark for human NeRF models with unified metrics and experimental settings, including training and testing ids (for generalisable methods), frame and view selection. Further, the work includes analysis of current human datasets and tries to provide a classification of testing data that would provide insights on the evaluated model. Further, the paper analyses generalisation ability of current methods, and the authors provide their new method for generalisable novel view rendering for human subjects.

---

> ### Author Response · Authors · 2023-08-22
>
> Thank your for your recognition of our work and valuable suggestions. The following are the initial designs based on your suggestions and clarifications for your confusions.
>
> **Q:** How train \& test views are chosen?
>
> **A:** We visualize the camera locations of dataset ZJU-MoCap,  as shown in https://github.com/nips-2023-paper-109/Human-Body-Rendering/blob/main/camera_vis.png, where the cameras of train views are uniformly located (the red cameras are for train views and the remaining are for test views). We also added the experimental comparisons when the 4 cameras of train views are located in neighbouring places, the results are shown in Tab. 3. When the selected train views are too close to each other, the performance of NeuralBody and AniNeRF drops severely.
>
> **Q:** Analysis on the poses in testing frames, i.e. are the poses in test split close to the ones in training.
>
> **A:** We will add this part following your suggestion.
>
> **Q:** A clear distinction of all settings should be put in the work.
>
> **A:** We will add such a table, and open-source the corresponding codes with proper adaptation.
>
> **Q:** Some more insights can be shared on the selection of hard clothing and hard poses for GeneBody dataset.
>
> **A:** We define hard clothing as loose, colorful clothing (especially skirts, traditional costumes), and hard poses as large-range, whole-body movement and rare poses. Currently, we can only classify these scenes by human inspection. But in the future, we will try to design an objective metric to classify automatically.
>
> **Q:** For completeness e.g. in supplementary, a masked PSNR could be used.
>
> **A:** We will add this part following your suggestion.
>
> **Q:** Does 4 correspond to balancing the number of training/test views in the dataset?
>
> **A:** In our design, single view means the monocular setting while four views represent a typical multi-view setting. Therefore, Tab. 2 in the main paper is to compare the performance of scene-specific methods under monocular and multi-view setting, respectively.
>
> **Q:** It would be good to show the performance with respect to position and viewing angle values to get a full overview.
>
> **A:** Fig. 1 of the main paper is for such a purpose, we will extend the figure by adding specific camera positions and viewing angle values.
>
> **Q:** It is a bit unclear which method is tested there (as well as in Table 8), as only the last line of the paragraph mentions the method (line 241). Also why was only one of 2 mentioned methods tested there?
>
> **A:** Sorry for the confusion, we selected the GP-NeRF as the representative method for generalzable methods, which we mentioned in line 185. We will also mention it in paragraph of 4.2 in the revised version and add more generalizable methods for this experiments.
>
> **Q:** It would be valuable to add training time as an information.
>
> **A:** We will add this part following your suggestion.
>
> **Q:** The description of the proposed GeneHumanNeRF is rather brief.
>
> **A:** We will add more descriptions of GeneHumanNeRF and open-source the corresponding codes for future study.
>
> **Q:** Describe how the benchmark would look like after the publication. How would the support for new models and datasets look like?
>
> **A:** We will release the first version of the codes for all settings of conducted experiments. Then, we will include as many existing human body rendering methods as possible to conduct the designed experiments. The results of all settings will finally be presented in a publicly available website. For new models and datasets, we will keep following on and include them into this benchmark.
>
> **Q:** Did you observe any correlation between HumanNeRF performance and the range of movements in the training examples.
>
> **A:** As the lbs weight field is overfitted with train data in HumanNeRF, its generalizability to novel pose might be limited. Therefore, with low coverage of movement around body axis, the lbs weights may be random output in the positions far from the body center, so the sampled points are warped to wrong places.
>
> **Q:** HumanNeRF after longer training provides more visually pleasing results. However, we notice significantly smaller PSNR. Do you have any qualitative observation of why that may happen. Is there a color shift between images or human translation not clearly visible with the naked eye?
>
> **A:** [1*] has a detailed analysis of why the metric of PSNR, SSIM sometimes may be inconsistent with LPIPS. Generally speaking, PSNR is a pixel-wise measurement while human's visual perception is hierarchical and context-dependent. And from our observations, a rendered image with high PNSR may still look blurred while a image with small LPIPS but low PSNR may look more realistic and reserve more sharpened edges.
>
>  [1*] Zhang, Richard, et al. "The unreasonable effectiveness of deep features as a perceptual metric." Proceedings of the IEEE conference on computer vision and pattern recognition. 2018.

---

> > ### Comment · Reviewer_1eYJ · 2023-08-29
> > **Response**
> >
> > Thank you for addressing my concerns. I appreciate the answers given and hope to see the promised changes in the revision.

---

### Official Review · Reviewer_dvPh · 2023-07-21
**(Response to author rebuttal at the end) Standardizing train & test splits for human NERF models**

**Rating:** 5
**Confidence:** 3
**Correctness:** What they describe seems correct.
**Clarity:** Generally OK.

**Strengths:**

The authors identify an important problem in the field -- it is certainly important that comparisons between different methods and papers are fair. The authors also recognize that cross-view and cross-subject generalizability are important features that have been under-explored in human NeRF models.

**Additional Feedback:**

Overall important work, although the clarity of the narrative, motivation, and novelty could be improved.


----rebuttal response

It looks like the authors have performed a considerable number of new experiments/analyses, thank you. However, I can't tell where or whether the revision contains the edited explanations that they mention. Also, I don't think they responded to my comment about the lack of a limitations section -- this is a glaring omission.

**Documentation:**

See opportunities for improvement.

**Ethics:**

No.

**Limitations:**

The authors do not have a limitations section.

**Opportunities For Improvement:**

The authors say they "introduce" the GeneBody and HuMMan datasets, which makes it sound like they are publishing these for the first time in this paper. This would certainly be a significant advance, but is this true that these are new? It appears as if these datasets have been published before.

Standardizing cropped regions and train/test splits for different datasets is certainly important, but I do wonder if NeurIPS is the best forum for such work without additional steps to validate the choices they have made. As it stands, the choices made in this paper seem somewhat arbitrary -- what happens if you change the "unification" strategy for evaluation? Should there perhaps be a composite metric across a range of choices? Such explorations would make the work here more "comprehensive" (borrowing the term from the title).

The authors should explain what makes their work "comprehensive" -- one could imagine testing many more variables and establishing many more metrics.

Perhaps the paper would benefit from a clearer summary, perhaps in a table, of all of their findings from their experiments and ablations. As it stands, findings are presented in a way where it is difficult to synthesize and overview of what has been found.

Their GeneHumanNeRF baseline should have more technical description in the main text, given the putative significance of the result. Furthermore, can the authors compare GeneHumanNeRF to other techniques? In the main text the only comparisons are internal, between different train/test datasets.

**Relation To Prior Work:**

Yes.

**Summary And Contributions:**

The authors seek to improve the objectivity and interpretability of human NeRF model evaluation by establishing more standardized evaluation metrics, such as normalized cropping and using specific train/test splits. In addition, they explore some of the limitations of existing methods and propose improvements to enhance generalizability and animatability.

---

> ### Author Response · Authors · 2023-08-22
>
> Thank for your valuable suggestions. we will take them to improve the proposed benchmark. The following are the initial designs based on your suggestions and clarifications for your confusions.
>
> **Q:** "introduce" the GeneBody and HuMMan datasets
>
> **A:** Sorry for the confusion, what we say "introduce" the GeneBody and HuMMan datasets means that we conduct experiments not only on the most commonly used dataset ZJU-MoCap but also on the GeneBody and HuMMan datasets to incorporate more diversity and challenging cases. We will modify the corresponding paragraph to avoid misunderstanding.
>
> **Q:** If NeurIPS is the best forum for such work without additional steps to validate the choices? The choices made in this paper seem somewhat arbitrary.
>
> **A:** In this paper, unifying different experiment choices is only a part of our work to address the issue of chaotic evaluation of existing works. Besides this part, we also investigated the effects of inaccurate human mask and SMPL parameters which are highly relied by existing human NeRF works, and analyzed the generalization ability and animatability of existing works. The experiment choices are made based on corresponding motivations. Specifically, the motivation of human cropping is to reduce the contribution of background area to the metrics calculation because the background area has already been masked out. The motivation of different train \& test view splits is to explore the performance difference under monocular (single view) setting and multi-view (four views) setting. The motivation of different train \& test frame split is to explore the effect of few train frames and many train frames. We have extend this part of experiment following the suggestion of Reviewer gjLn.
>
> **Q:** Composite metric across a range of choices
>
> **A:** We will try to design a new metric that will consider the frame consistency and other experimental settings.
>
> **Q:** What makes the work "comprehensive" ?
>
> **A:** As mentioned above, we unified the existing chaotic evaluation settings,also investigated the effects of inaccurate human mask and SMPL parameters, and analyzed the generalization ability and animatability of existing works and proposed a simple baseline of a useful but under-explored setting, i.e. animatable and generalizable human NeRF from monocular videos.
>
> **Q:** Perhaps the paper would benefit from a clearer summary, perhaps in a table, of all of their findings from their experiments and ablations.
>
> **A:** we will add this part following your suggestion.
>
> **Q:** GeneHumanNeRF baseline should have more technical description in the main text, and compare GeneHumanNeRF to other techniques.
>
> **A:** We will add more descriptions for the proposed baseline GeneHumanNeRF. As this new baseline is a method of new setting, there is currently no other existing comparable method.

---

> ### Author Response · Authors · 2023-08-31
>
> Dear Reviewer,
>
> We want to express our sincere gratitude for your insightful suggestions which are instrumental in enhancing the quality of our work. We hope that our proposed modifications would have addressed your concerns about the clarity of our presentation. We would really appreciate it if you could let us know if there are any further questions or aspects of the paper that require additional clarification.
>
> Thank you once again for your time and consideration.

---

### Official Review · Reviewer_a43V · 2023-07-23

**Rating:** 7
**Confidence:** 3
**Correctness:** Yes
**Clarity:** Yes

**Strengths:**

> + Comprehensive summary of existing human NeRF methods: The summary in Table 1 is clear and informative.
> + Beneficial Analysis and Observations: The summarized observations in Sec. 4 can be helpful for further works on Human NeRF and save others' efforts on tuning the baselines.
> + Well-designed experiments: The experiments are designed to include sufficient metrics and challenging and varying datasets.

**Additional Feedback:**

N/A

**Documentation:**

Yes

**Ethics:**

No ethical concern

**Limitations:**

Yes

**Opportunities For Improvement:**

> + Why benchmark for Human NeRF: The motivation for building the benchmark for human NeRF is not clear. What are the key differences between other NeRF works and Human NeRF works? Can benchmark for other NeRF works also be used for Human NeRF? Are the observations consistent for other NeRF works and Human NeRF?

> + Justification of experiments settings (e.g., hyperparams): some experiments are missing and lack justification. For example, for the part of "Upper-bound performance of finetuning on 244 a single subject", there is no description of what the training recipe used for 15 minutes, 1 hour, and 20 hours settings. How to justify whether the corresponding experiment settings are fair for the comparison?

**Relation To Prior Work:**

Yes

**Summary And Contributions:**

This work introduces a benchmark for human NeRF models, which is pre-trained on large-scale datasets to assess generalizable methods. The contributions can be summarized as: (1) establishing a comprehensive benchmark for human NeRF models, (2) including challenging datasets, (3) providing generalizable models trained on large-scale datasets, and (4) analyzing the key elements of either animatability or generalizability.

---

> ### Author Response · Authors · 2023-08-22
>
> Thank you for your recognition of our work. We clarify your confusions in the following.
>
> **Q:** Why benchmark for Human NeRF?
>
> **A:** The human-centered NeRF works progress rapidly. However, there is still no unified evaluation settings (e.g., used views, frames, ground-truth masks, and SMPLs), as shown in Tab. 1 of the main paper. Secondly, the most commonly used dataset ZJU-MoCap is oversimplified, which cannot reflect the real performance of existing methods. Thirdly, the generalizability and animatability has rarely been systematically compared and discussed in existing works although they are important aspects of human NeRFs.
>
> **Q:** What are the key differences between other NeRF works and Human NeRF works?
>
> **A:** Typical class-agnostic NeRF works like vanilla NeRF and InstantNGP can only model static scene/objects while human NeRF works are able to model dynamic humans with the help of human body priors. Thus, to acquire high-quality novel view renderings, class-agnostic NeRF works require a batch of multi-view images for training while Human NeRF works can take sparse-view or even single-view image sequences for training. Furthermore, human NeRF works can render a human of unseen poses assigned by users.
>
> **Q:** Can benchmark for other NeRF works also be used for Human NeRF? Are the observations consistent for other NeRF works and Human NeRF?
>
> **A:** Benchmark for other NeRF works can only be a reference but is not applicable to Human NeRF works as they focus more on modeling dynamic human and taking the advantages of human body priors. For example, only human NeRF works can measure the performance of novel pose rendering and analyze the effect of inaccurate SMPL parameters.
>
> **Q:** Justification of experiments settings.
>
> **A:** Sorry for the confusions. In the experiment for upper-bound performance evaluation, we select the representative generalizable method GP-NeRF and representative scene-specific method HumanNeRF to be finetuned/trained in controlled time, i.e. 15 minutes, 1 hour and 20 hours. We use the default settings of them as we want to compare the convergence speed and upper-bound performance with enough training time. Here, 15 minutes represent a relatively short training time while 20 hours represent a relatively long training time. Similarly, in the experiment of train views (Tab. 2 in the main paper), the single view represents the monocular setting while the four view represents a commonly used multi-view setting.

---

> ### Author Response · Authors · 2023-08-31
>
> Dear Reviewer,
>
> We want to express our sincere gratitude for your insightful suggestions which are instrumental in enhancing the quality of our work. We hope that our proposed modifications would have addressed your concerns about the clarity of our presentation. We would really appreciate it if you could let us know if there are any further questions or aspects of the paper that require additional clarification.
>
> Thank you once again for your time and consideration.

---

### Official Review · Reviewer_gjLn · 2023-07-27
**+Good motivation and attempt； -Lack in-depth analysis, sota methods;**

**Rating:** 6
**Confidence:** 5
**Clarity:** In general, it is easy to understand.

**Strengths:**

+ I'm glad that the authors noticed the unfair/chaotic comparisons existing in each individual work, and tried to align the criterion. Since it is important for the community to not be misled by the phenomena observed from unfair/biased comparisons.

+ The authors tried to unfold the data-preprocessing factors that influence the performance via both quantitative and qualitative demonstrations.

+ A modified HumanNeRF baseline is presented.

**Additional Feedback:**

Please refer to the above.

Overall, I acknowledge the attempt of this work to present unified human body rendering benchmarks. But, I hope to see improvements. What prevents me to lean towards an acceptance is mainly about the inadequate benchmark designs and analysis. I'd like to raise the rating if the concerns are dispelled with more results and discussion during the rebuttal.

**Correctness:**

There are some  inaccurate claims:
- Tab1 GNR 3*(views)->4*
- line106  zju 23 views -> 24 views or you could clarify some views are missing/invalid in practice, and most/avg of views are 23
- l112 Humman claims 1000 in the original paper, and releases 153 in practice. Where the number 300 comes from?
- l132-134  It seems like there is no causality between the sentences.
- l140, for the 16 views, which dataset you refer to? Or it is just an example?
- l176-l77,  I kinda doubt the conclusion, as inaccuracy smplx would lead to correspondence misaligned between the two spaces.

Other comments:
-  Animatability.  It is understandable, and some media use the term, but not a good one.
-  line 29 They ->It
- line 52 subject -> subjects, hard clothes -> challenging clothes.
-line 89 *into their first* of what?

**Documentation:**

N/A

**Ethics:**

I didn't see obvious ethical concerns on the current submission.

**Limitations:**

There is no significant negative societal impact in the current submission.

**Opportunities For Improvement:**

My main concerns are the inadequate benchmark designs （methods/settings/metrics）and analysis. Please refer to the following comments.

+ The authors noticed the insufficient/easy sample problems in ZJU and tried to benchmark on more challenging ones. However, in practice, the main evaluations still lie on ZJU. This might lead to several biased conclusion problems -- (1)Traning frames gap (Sec3.3). ZJU contains repeat motion in a sequence. It would be better if the authors could conduct a more detailed ablation regarding frame sampling (different sampling strategies, the trends (not only 60/300, but more frame setting))  and motion repeat, Also, this aspect extends another problem -- these datasets have different modalities. For example, some samples in Genebody 1.0 repeat their actions toward different cameras (e.g., fuzhizhi). So, selecting different view locations might influence the performance conclusions, and there might be an intertwined relationship between frames and views.  The authors should consider these inhere modality differences when designing experimental settings.  And please also clarify how you choose the 1/4 views in line126, and how the four views were distributed .
(2) SMPL/Mask influence discussion (Sec 3.4 ). Given ZJU only includes easy motion and close-fitting clothes, the fitting and masking errors are lower than other challenging datasets  (even using the same estimators) with complicit motions/extreme views/loosened clothes. The different error distributions/error types might lead to different conclusions. Thus, the authors are expected to unfold sec3.4 by (a) evaluating the other two datasets, and (b) discussing the annotation errors under different error type scenarios (Fig2 basically the same type - slimmer shape and shifted global orientation.)  (c)<better have> It would be interesting to see how different annotation tools provided by these datasets would influence the results on the same data samples. (d) It is expected to first show the accuracy (e.g., MPJPE)  of GT and HybrIK, and then unfold the discussion of this section. (3) Tab 8. (a) Do you apply color correction between the two datasets? Since the two datasets have different color consistency degrees. (b) Different subject samples might lead to bias. Please clarify how you choose the human test samples, and what if the samples are changed to different ones with different motion/appearance distribution? (c) What if train on Humman, and test on Genebody? Will it lead to a different conclusion? Also,  the methodologies of different methods might lead to different conclusions too.  The authors should also include other generalized methods (such as NHP/GNR/keypointnerf, or case-agnostic methods).

+ As a benchmark work, it is expected to include more sota methods with different methodologies. So that the readers/community could sense the direction of future improvements. However, the current submission lacks representative methods like NeuMan ([monocular video setting] open-sourced since last year: https://github.com/apple/ml-neuman) , and Neural Actor( https://github.com/lingjie0206/Neural_Actor_Main_Code). Also, it would be interesting to see how class-agnostic methods (e.g., pixel nerf/Instant-NGP) compete with the ones with human body priors.

+ Evaluation metrics could be better designed(I did not include this in my rating, just a suggestion). Given current used metrics can only evaluate every single frame, it would be better if the authors could introduce some VQA methods (e.g, VFMA used in Actors-HQ) or other alternatives that consider the consistency/reasonability among frames. I'd like to see new observations and contribution on this aspect.

+ Instead of just testing several SOTA methods (Page 7), I would strongly suggest clarifying what are the detailed design/methodology differences in each method, and how they make influences on the results under different scenarios.

+ For the proposed modified baseline, it is expected to compare with MPS-NeRF to see the performance gap. Since the modification is inspired by MPS-NeRF.

**Relation To Prior Work:**

Yes

**Summary And Contributions:**

This paper presents several benchmarks upon ZJU-MoCap/Humman/Genebody1.0, with an attempt to unify the criterion of human body rendering evaluations. Meanwhile, the authors also provide a modified HumanNeRF, which extends the original monocular per-case setting (Weng et al CVPR2022) to a generalized one.

---

> ### Author Response · Authors · 2023-08-22
>
> Thank you for your valuable suggestions, we will take them to improve the proposed benchmark. The following are the initial designs based on your suggestions and clarifications for your confusions.
>
> **Q:** More ablation study of train frame number and frame interval.
>
> **A:** We conducted more experiments with different train frame numbers, as shown in Tab. 1. Then, we fix the train frame to be 60, and sample train frames with different intervals. The results are shown in Tab. 2.
>
> **Q:** Influence of selected view locations.
>
> **A:** The number of train view to be 1 means the monocular setting while 4 denotes a typical multi-view setting. We visualized camera locations in https://github.com/nips-2023-paper-109/Human-Body-Rendering/blob/main/camera_vis.png, where the cameras of train views are uniformly located (the red cameras are for train views and the remaining are for test views). We also added the experimental comparisons when the 4 cameras of train views are located in neighbouring places, the results are shown in Tab. 3.  When the selected train views are too close to each other, the performance of NeuralBody and AniNeRF drops severely.
>
> **Q:** Influence of inaccurate SMPL/Mask on more challenging dataset.
>
> **A:** We will conduct more experiments on both GeneBody and HuMMan. In addition, we have calculated the accuracy of the estimated SMPL parameters of ZJU-MoCap dataset in the metrics of MPJPE and PA-MPJPE, which are 69.94 mm and 51.33 mm respectively.
>
> **Q:** Do you apply color correction between the two datasets?
>
> **A:** No, we did not do any preprocessing steps.
>
> **Q:** Please clarify how you choose the human test samples, and what if the samples are changed to different ones with different motion/appearance distribution?
>
> **A:** For HuMMan dataset, we follow its suggested train/test split. For ZJU-MoCap dataset, we follow the train/test split of NHP. By human inspection, we have not found obvious motion/appearance imbalance in these split schemes.
>
> **Q:** More comparisons of cross-dataset generalization.
>
> **A:** We added the setting of training on HuMMan and testing on GeneBody. The results of GP-NeRF are shown in Tab. 4. We will then add NHP and KeypointNeRF for this part of experiments.
>
> **Q:** Benchmark results with more recent methods.
>
> **A:** We will keep updating new methods and datasets into this benchmark, including other existing published methods. For the suggested methods NeuMan and Neural Actor, they cannot directly run on ZJU-MoCap dataset as they requires more preprocessed materials, so it may take some time to include them into the benchmark.
>
> **Q:** Human-specific methods V.S. Class-agnostic methods
>
> **A:** Class-agnostic methods can only model static scene/objects while human-specific methods can model dynamic humans with body priors. We set the train frame to be 1 (so that the human can be viewed as static), and use four views for training. We choose InstantNGP as the representative class-agnostic method to conduct experiments. We found that InstantNGP can overfit well to the train views but collapse severely for test views. We conjecture this is because the number of views are too few for class-agnostic methods.
>
> **Q:** Discussion of methodology differences and their impact.
>
> **A:** We will plot new figures to present their methodology differences and analyze the impact of different modules.
>
> **Q:** Evaluation metrics could be better designed.
>
> **A:** We will try to design a new metric that will consider the frame consistency.
>
> **Q:** Comparison with MPS-NeRF.
>
> **A:** Our proposed baseline is a new setting, i.e. animatable and generalizable human NeRF from monocular video, which is different from MPS-NeRF whose input is multi-view images. Therefore, the results of the two methods are not comparable.
>
> **Q:** l112 Humman claims 1000 in the original paper, and releases 153 in practice. Where the number 300 comes from?
>
> **A:** Sorry for the confusion, 153 is the number of identities, and each identity may have 1-3 multi-view video sequences. So, the number 300 refers to the total number of multi-view video sequences. We will correct the description in the revised version.
>
> **Q:** l140, for the 16 views, which dataset you refer to? Or it is just an example?
>
> **A:** It is just an example to describe the difference of novel view rendering and novel pose rendering. We will change it to the realistic number of ZJU-MoCap dataset to avoid misunderstanding.
>
> **Q:** l176-l77, I kinda doubt the conclusion, as inaccuracy smplx would lead to correspondence misaligned between the two spaces.
>
> **A:** Inaccuracy SMPL parameters do hurt the performance of NeuralBody, NHP and GP-NeRF severely, but empirically HumanNeRF drops its quantitative performance slightly, as shown in Tab. 6 of the main paper. And the qualitative results are also consistent with this conclusion, as shown in Fig. 2 of the main paper. We will try to conduct more experiments to figure out the behind reason.

---

> ### Author Response · Authors · 2023-08-31
>
> Dear Reviewer,
>
> We want to express our sincere gratitude for your insightful suggestions which are instrumental in enhancing the quality of our work. We hope that our proposed modifications would have addressed your concerns about the clarity of our presentation. We would really appreciate it if you could let us know if there are any further questions or aspects of the paper that require additional clarification.
>
> Thank you once again for your time and consideration.

---

### Author Response · Authors · 2023-08-22
**General Response**

We sincerely thank the reviewers for their valuable and insightful feedback. We are pleased that the reviewers found our work well-motivated (gjLn), helpful (a43V), important (dvPh, 1eYJ) and interesting (1eYJ).
In the revised version,we have made the following improvements:
* We improved the clarity of some sentences and definitions as suggested. (gjLn, a43V, dvPh, 1eYJ)
* We designed more experiments in the aspect of train frame numbers, train frame intervals, train view selections. (gjLn, 1eYJ)
* We clarified the behind motivations of each experimental setting, and visualized how we select camera views and analyzed its impact. (gjLn, a43V, dvPh, 1eYJ)
* We added the computed accuracy of estimated SMPL parameters in the metrics of both MPJPE and PA-MPJPE. In addition, we also conducted experiments of inaccurate SMPL parameters on the datasets of GeneBody and HuMMan. (gjLn)
* We added more experiments for the cross-dataset generalization, including train on HuMMan then evaluate on GeneBody, and added more generalizable methods to conduct experiments. (gjLn)
* We added more analysis of the methodology differences among representative methods and how they influence the performance in different situations. (gjLn)
* We added more analysis of novel pose rendering in the aspect of similar pose percentage. (1eYJ)
* We added more description and analysis of how to distinct hard clothing and hard poses. (1eYJ)
* We added the experiments of training time ablation. (1eYJ)
* We added more details to introduce our proposed baseline GeneHumanNeRF. (gjLn, dvPh, 1eYJ)

Due to the time and computational resource limit, some of the above experiments are still in progress, we will try our best to finish all experiments and present the results as soon as possible. We appreciate all the suggestions made by reviewers to improve our work. We are looking forward to further feedback.

Table 1:  Quantitative comparison of representative scene-specific methods with different numbers of train
frames on ZJU-MoCap. The train views is fixed to be 4.
|  |  1frame |  60 frames | 100 frames | 300 frames | 500 frames |
| :--- |    :---  |  :--- |  :--- |  :--- |  :--- |
|Methods |PSNR ↑ SSIM ↑ LPIPS ↓| PSNR ↑ SSIM ↑ LPIPS ↓ |PSNR ↑ SSIM ↑ LPIPS ↓ |PSNR ↑ SSIM ↑ LPIPS ↓ |PSNR ↑ SSIM ↑ LPIPS ↓|
| NeuralBody | 23.83 0.8881 0.1412 |26.29 0.9248 0.1087| 27.35 0.9364 0.0974| 27.82 0.9396 0.1048 |27.77 0.9380 0.1137|
| AniNeRF | 24.56 0.8987 0.1298| 24.45 0.8899 0.1652| 24.78 0.8959 0.1558 |24.13 0.8822 0.1867 |24.39 0.8865 0.1882|
| HumanNeRF | 25.98 0.9031 0.1015| 26.49 0.9225 0.0800| 26.40 0.9253 0.0730| 25.86 0.9246 0.0728| 25.52 0.9218 0.0754|

Table 2: Quantitative comparison of representative scene-specific methods with different numbers of train frame
sampling intervals on ZJU-MoCap. The number of train views is fixed to be 4 and the train frames to be 60.
|  |  interval 1 | interval 5 | interval 10 |
| :--- |    :---  |  :--- |  :--- |
|Methods| PSNR ↑ SSIM ↑ LPIPS ↓| PSNR ↑ SSIM ↑ LPIPS ↓| PSNR ↑ SSIM ↑ LPIPS ↓|
|NeuralBody| 26.29 0.9248 0.1087| 27.75 0.9394 0.0947 |27.78 0.9393 0.0967|
|AniNeRF |24.45 0.8899 0.1652| 24.54 0.8872 0.1746 |25.12 0.8991 0.1695|
|HumanNeRF| 26.49 0.9225 0.0800 |25.84 0.9251 0.0716| 25.75 0.9249 0.0745|

Table 3: Quantitative comparison of representative scene-specific methods with different train view distribution
on ZJU-MoCap. The train views is fixed to be 4 and the train frames to
be 300.
|  |  uniform | half-uniform | quarter-uniform | neighbouring |
| :--- |    :---  |  :--- |  :--- |  :--- |
|Methods |PSNR ↑ SSIM ↑ LPIPS ↓ |PSNR ↑ SSIM ↑ LPIPS ↓| PSNR ↑ SSIM ↑ LPIPS ↓| PSNR ↑ SSIM ↑ LPIPS ↓|
|NeuralBody| 27.82 0.9395 0.1048| 27.18 0.9339 0.1084 |  25.96  0.9190 0.1213 | 21.72 0.8590 0.1823|
|AniNeRF |24.12 0.8822 0.1867| 23.82 0.8792 0.1902 | 23.09 0.8687 0.2010 |21.24 0.8416 0.2272|
|HumanNeRF| 25.86 0.9246 0.0727| 25.74 0.9227 0.0749 | 25.09 0.9142 0.0869 |24.03 0.8958 0.1116|

Table 4: Quantitative comparison of cross-subjects
generalization of generalizable methods.GB means
GeneBody dataset.

|Train set |Test set |PSNR ↑ SSIM ↑ LPIPS ↓|
| :--- |    :---  |  :--- |
|HuMMan| HuMMan Eval| 19.66 0.8593 0.2837|
|HuMMan| ZJU-MoCap| 23.82 0.8810 0.1850|
|HuMMan| GB Normal| 18.88 0.8125 0.3995|
|HuMMan| GB Hard Cloth| 15.06 0.7277 0.3853|
|HuMMan| GB Hard Pose| 16.56 0.7672 0.3391|

Table 5: Quantitative comparison of representative scene-specific methods with different train time
on ZJU-MoCap.

|  |  1 hour | 5 hours | 10 hours | 20 hours |
| :--- |    :---  |  :--- |  :--- |  :--- |
|Methods |PSNR ↑ SSIM ↑ LPIPS ↓ |PSNR ↑ SSIM ↑ LPIPS ↓| PSNR ↑ SSIM ↑ LPIPS ↓| PSNR ↑ SSIM ↑ LPIPS ↓|
|NeuralBody| 27.70 0.9374 0.1180 | 27.88 0.9402 0.0998 |  27.86  0.9404 0.0947 | 27.84 0.9403 0.0934 |
|AniNeRF |24.17 0.8810 0.1909 | 24.06 0.8830 0.1845 | 23.92 0.8837 0.1829 |23.87 0.8838 0.1825 |
|HumanNeRF| 23.77 0.8900 0.1191 | 24.97 0.9137 0.0871 | 25.52 0.9210 0.0778 | 25.85 0.9252 0.0729 |

---

### Decision · Program_Chairs · 2023-09-22

**Decision:**

Accept (Poster)

**Comment:**

The paper addresses the problem of a comprehensive benchmark for neural human body rendering. The authors argue that the current tremendous research interest in the topic has generated a variety of methods and diverse datasets, which makes a proper evaluation challenging. To address this gap they propose to design and execute a number of studies using unified evaluation settings and various metrics to establish a fair and comparable benchmark for human NeRF models. The paper also describes a cross-subject benchmark to assess generalisability of the methods. Finally, the authors propose how to make HumanNeRF generalizable as a benchmark baseline. The initial assessments of the quality of the paper by all four reviewers have mostly been consistent. While they have positively commented on the timeliness and importance of the proposed benchmark, they have also raised several issues related to the execution and thoroughness of the design of the benchmark and experiments. They requested several clarifications and proposed numerous directions for improvements (e.g., better motivation, narrative, more in-depth insights in the design choices made, several new experiments related to humans in different clothes and poses, etc.). In response, the authors provided the requested clarifications and designed and executed several new experiments. Most of the reviewers acknowledged these efforts by raising their scores above the acceptance bar. However, since some of the additional experiments are still in progress, the authors are urged to do their best to complete the final version of the manuscript with all the improvements that were promised during the rebuttal stage.